# Optical Conductivity as a Probe of the Interaction-Driven Metal in Rhombohedral Trilayer Graphene

**DOI:** 10.3390/nano12213727

**Published:** 2022-10-24

**Authors:** Vladimir Juričić, Enrique Muñoz, Rodrigo Soto-Garrido

**Affiliations:** 1Departamento de Física, Universidad Técnica Federico Santa María, Casilla 110, Valparaíso 2340000, Chile; 2Nordita, KTH Royal Institute of Technology and Stockholm University, Hannes Alfvéns väg 12, 106 91 Stockholm, Sweden; 3Facultad de Física, Pontificia Universidad Católica de Chile, Vicuña Mackenna 4860, Santiago 8331150, Chile

**Keywords:** trilayer graphene, optical conductivity, electron-electron interactions

## Abstract

Study of the strongly correlated states in van der Waals heterostructures is one of the central topics in modern condensed matter physics. Among these, the rhombohedral trilayer graphene (RTG) occupies a prominent place since it hosts a variety of interaction-driven phases, with the metallic ones yielding exotic superconducting orders upon doping. Motivated by these experimental findings, we show within the framework of the low-energy Dirac theory that the optical conductivity can distinguish different candidates for a paramagnetic metallic ground state in this system. In particular, this observable shows a single peak in the fully gapped valence-bond state. On the other hand, the bond-current state features two pronounced peaks in the optical conductivity as the probing frequency increases. Finally, the rotational symmetry breaking charge-density wave exhibits a minimal conductivity with the value independent of the amplitude of the order parameter, which corresponds precisely to the splitting of the two cubic nodal points at the two valleys into two triplets of the band touching points featuring linearly dispersing quasiparticles. These features represent the smoking gun signatures of different candidate order parameters for the paramagnetic metallic ground state, which should motivate further experimental studies of the RTG.

## 1. Introduction

Quasi-two-dimensional graphene-based van der Waals (vdW) heterostructures, such as bilayer and trilayer graphene, have recently emerged as a groundbreaking territory for the discovery of new electronic states of quantum matter driven by the electron interactions [1,2,3,4,5,6,7,8,9,10,11,12]. It is rather remarkable that by externally tuning the twist angle, doping, and/or the magnetic field, a new landscape of exotic insulating, metallic and superconducting states has been unearthed in these systems. Particularly prominent in this respect is the interplay between the metallic and the superconducting phases that gives rise to very rich phase diagrams (for a recent review, see Ref. [13]). However, their theoretical understanding is often hampered by the difficulty in distinguishing possible candidate ground states in these systems, as, for instance, when considering the emergence of superconductivity from a parent metallic state.

Rhombohedral trilayer graphene (RTG) has recently emerged as a rather prominent example in this respect, where superconducting instabilities are in proximity to metallic ground states in different doping regimes [11,12], with a few theoretical scenarios proposed to explain the rich phenomenology [14,15,16,17,18,19,20,21]. Most interestingly, very little is known about an exotic metal in the proximity of the experimentally identified SC1 order [12], except that it exhibits paramagnetic nature (see Extended Figure 7 in Ref. [12] and the discussion therein). In fact, a few candidates for such a state that may be driven by electron interactions have been identified, each of them breaking different microscopic symmetries: the valence-bond order (VBO), bond-current order (BCO), and smectic charge-density wave (sCDW) orders [16], respectively.

The optical conductivity is a well established tool in studying correlated electron materials, which is directly related to the excitation spectrum [22]. In particular, it has been studied in various vdW materials both theoretically [23,24,25,26,27,28] and experimentally [29,30]. In this work, we focus on the collisionless or high-frequency regime of the optical conductivity, pertaining to the frequencies $\hbar \omega \gg k_{B}T$, since in this regime, this observable shows a universal scaling that depends only on the form of the dispersion of the low-energy quasiparticles, the dimensionality of the system, and the scaling dimension of the electron–electron interactions [31]. Importantly, the scaling dimension of the optical conductivity in a *d*-dimensional system is equal to d−2 in units of momentum. Therefore, exactly in d=2, at finite frequency and temperature, with other energy scales set to zero, $\sigma \left(\omega ,T\right)=\left(e^{2}/h\right)f\left(\hbar \omega /k_{B}T\right)$, with f(x) as a universal dimensionless scaling function. In the limit x→∞, this function tends to a constant, yielding a universal amplitude for the collisionless optical conductivity.

Motivated by these developments, we show that the collisionless optical conductivity can distinguish different candidate paramagnetic metallic ground states in RTG. In particular, this observable shows a single peak in the fully gapped valence-bond state, as displayed in Figure 1a. On the other hand, the bond-current state features two pronounced peaks in the optical conductivity as the probing frequency increases, see Figure 1b. This behavior can be directly related to the Dirac nature of the valence-bond and the bond-current order parameters obeying, respectively, anticommutation and commutation relations with the single-particle noninteracting Hamiltonian given by Equation (Equation 1), and with the behavior of the density of states (DOS); see Figure 2a,b. Finally, the sCDW state is characterized by a minimal conductivity, which is independent of the amplitude of the sCDW order parameter [Figure 1c], and it corresponds precisely to the splitting of the two cubic nodal points at the two valleys into two triplets of linearly dispersing band touching points (Figure 3). This behavior is also consistent with the form of the DOS, particularly at low energies, displaying the linear scaling with the energy [Figure 2c]. As such, these characteristic features in the optical conductivity represent the smoking gun signatures of different candidate order parameters for the paramagnetic metallic ground state of the RTG. These, in turn, can serve as a starting point for the study of the superconductivity in this system and should motivate further its experimental study.

## 2. Model

We consider the effective low-energy model for the rhombohedral (ABC stacked) trilayer graphene obtained after integrating out high-energy degrees of freedom corresponding to the four gapped bands consisting of the states at the dimerized sites (one in the bottom (*A*), two in the middle (*B*), and one in the top (*C*) layer) [32,33,34]. Taking into account the layer (or equivalently sublattice) and valley degrees of freedom, the single-particle Hamiltonian for noninteracting electrons reads
(1)$$\begin{array}{c} H_{0}=\alpha \left[f_{1}\left(k\right)\Gamma_{31}+f_{2}\left(k\right)\Gamma_{02}\right]+u\Gamma_{03}-\mu \Gamma_{00} \end{array}$$
where $\alpha =t_{0}^{3}a^{3}/t_{⊥}$, a≃0.25 nm is the lattice spacing within the single graphene layer, while $t_{0}≃2.5$ eV and $t_{⊥}≃0.5$ eV are the intralayer and the interlayer nearest-neighbor hoppings, respectively [33]. The corresponding bandwidth is given by the cutoff scale for the low-energy model in Equation (Equation 1), *t*∼$t_{⊥}$∼0.5 eV. The form factors $f_{1}\left(k\right)=k_{x}\left(k_{x}^{2}-3k_{y}^{2}\right)$, $f_{2}\left(k\right)=-k_{y}\left(k_{y}^{2}-3k_{x}^{2}\right)$, respectively, transform under $A_{1u}$ and $A_{2u}$ representations of the $D_{3d}$ point group of the RTG. Momentum k is measured from the respective band-touching points (valleys). Electron (hole) doping corresponds to μ>0(μ<0), and we set $\hbar =k_{B}=e=1$ hereafter. The four-dimensional matrices are $\Gamma_{\mu \nu }=\tau_{\mu }\sigma_{\nu }$, where $\left\{\tau_{\mu }\right\}$ and $\left\{\sigma_{\nu }\right\}$ are the sets of Pauli matrices that act on the valley and sublattice (layer) indices, respectively. Since we here consider only paramagnetic metallic states, we suppress the spin indices, see also Appendix A of the Appendix A. Taking the possible patterns of symmetry breaking and the low-energy degrees of freedom, three candidates emerge in this respect [16]. First of all, the VBO, which fully gaps the system out and breaks the sublattice symmetry. Second, the BCO breaks, besides the sublattice symmetry, as well as the time-reversal. Finally, the sCDW breaks both the U(1) rotational symmetry about the z− axis, generated by the matrix $\Gamma_{33}$, down to the discrete $C_{3}$ subgroup, and the lattice-translational symmetry, generated by the matrix $\Gamma_{30}$. These three orders are represented by the following matrices (ρ=1,2):
(2)$$\Gamma_{\rho 1}\;\left(\mathrm{VBO}\right),\;\Gamma_{\rho 2}\;\left(\mathrm{BCO}\right),\;\left(\Gamma_{\rho 0},\Gamma_{\rho 3}\right)\;\left(\mathrm{sCDW}\right).$$

The corresponding irreducible representations of the $D_{3}$ group are $A_{1}$ and $A_{2}$, for the VBO and BCO, respectively, while the sCDW transforms under the two-dimensional *E* representation. Notice that we now reduce the symmetry down to $D_{3}$ subgroup of the full $D_{3d}$ point group of the noninteracting low-energy Hamiltonian in Equation (Equation 1) because we allow for the backscattering processes that mix the valleys, and in turn may yield the translational-symmetry breaking orders, such as the sCDW.

## 3. Optical Conductivity

To distinguish different candidate paramagnetic metallic ground states in RTG, we now compute the optical conductivity. To this end, we use the Kubo formula for the linear optical conductivity [35]
(3)$$\sigma_{ij}\left(\Omega \right)=\lim_{i\Omega_{n}\to \Omega +i0^{+}}\frac{i\Pi_{ij}\left(i\Omega_{n}\right)}{\Omega },$$
where the polarization tensor reads
(4)$$\begin{array}{cc} \Pi_{ij}\left(i\Omega_{n}\right)= & -T\sum_{n}\int \frac{d^{2}k}{{\left(2\pi \right)}^{2}}\mathrm{Tr}\left[{\hat{v}}_{i}G\left(i\omega_{n}+i\Omega_{n},k\right)\right.\\ \; & \left.\times {\hat{v}}_{j}G\left(i\omega_{n},k\right)\right]. \end{array}$$

Here, the velocity operator ${\hat{v}}_{i}=\partial H/\partial k_{i}$, $G\left(i\omega_{n},k\right)={\left[i\omega_{n}-H\right]}^{-1}$ is the Matsubara Green’s function, $\omega_{n}=\left(2n+1\right)\pi T$ are the fermionic Matsubara frequencies, and the analytical continuation onto real frequencies is performed, $i\Omega_{n}\to \Omega +i0^{+}$, see also Appendix A for details.

## 4. Valence-Bond Order

We start by computing the optical conductivity for the simplest case, the VBO. The Hamiltonian is given by $H=H_{0}+\Delta_{1}\Gamma_{11}+\Delta_{2}\Gamma_{21}$. Notice that since time-reversal symmetry (TRS) is preserved, the xy component of the polarization tensor vanishes. In addition, since rotational symmetry is conserved, $\Pi_{xx}=\Pi_{yy}$, so we need to compute only one of the polarization tensor components. We then find, with the details shown in the Appendix A,
(5)$$\begin{array}{cc} \Pi_{xx}\left(\Omega \right)= & \;72\alpha^{2}\int \frac{d^{2}k}{{\left(2\pi \right)}^{2}}\frac{k^{4}\left(u^{2}+\Delta^{2}+\alpha^{2}k^{4}k_{x}^{2}\right)}{E\left(k\right)\left[4E^{2}\left(k\right)+\Omega_{n}^{2}\right]}\\ \; & \times \Theta \left(\Omega -2\sqrt{u^{2}+\Delta^{2}}\right)\delta n_{F}\left(E\right), \end{array}$$
where $E\left(k\right)=\sqrt{\alpha^{2}k^{6}+u^{2}+\Delta^{2}}$ is the quasiparticle dispersion, $n_{F}\left(z\right)={\left(e^{z/T}+1\right)}^{-1}$ is the Fermi–Dirac distribution, and we defined $\delta n_{F}\left(E\right)=n_{F}\left(-E-\mu \right)-n_{F}\left(E-\mu \right)$. After performing the analytic continuation to real frequency, $i\Omega_{n}\to \Omega +i0^{+}$, we obtain the real part for the optical conductivity,
(6)Reσxx(Ω)=−ImΠxx(Ω)Ω=σ0Ω2+4(u2+Δ2)Ω2δnF|Ω|/2×Θ(Ω−2u2+Δ2),
where $\sigma_{0}=3/8$ is the universal optical conductivity for the noninteracting spinless RTG in the collisionless regime (in units $e^{2}/\hbar$), and we defined $\delta n_{F}\left(E\right)=n_{F}\left(-E-\mu \right)-n_{F}\left(E-\mu \right)$. The technical details are presented in Appendix A.

In Figure 1a, we can see a plot for the optical conductivity at zero temperature and at the neutrality point μ=0. Notice that the optical conductivity is zero until $\Omega \geq 2\sqrt{u^{2}+\Delta^{2}}$, where there is a maximum. This is expected, since $2\sqrt{u^{2}+\Delta^{2}}$ is the gap between the conducting and valence bands. The asymptotic value is given by Reeσxx(Ω)→σ0 when $\Omega \gg \sqrt{u^{2}+\Delta^{2}}$.

To further elucidate the origin of the peaks in the optical conductivity in the VBO, we analyze the corresponding DOS
(7)$$N_{VBO}\left(E\right)=\frac{\left|E\right|{\left(E^{2}-u^{2}-\Delta^{2}\right)}^{-\frac{2}{3}}}{6\pi \alpha^{2/3}}\Theta (|E|-\sqrt{u^{2}+\Delta^{2}}).$$

A remarkable feature of this expression is that the DOS develops a pole at the minimum of the conduction band $E=\sqrt{u^{2}+\Delta^{2}}$, and hence it grows as compared to the gapless case u=Δ=0. This effect manifests in an enhancement of the optical conductivity with the gap as compared to the universal value $\sigma_{0}$ for the gapless case, as directly follows from Equation (Equation 6).

## 5. Bond-Current Order

We now follow the same procedure for the BCO case, where the Hamiltonian is given by $H=H_{0}+\Delta_{1}\Gamma_{12}+\Delta_{2}\Gamma_{22}$. As for the VBO case, the rotation symmetry is preserved in the BCO, so $\Pi_{xx}=\Pi_{yy}$. The real part of the optical conductivity then reads
(8)Reeσxx(Ω)=32Ω2Θ|Ω|−2uΘ2u2+Δ2−|Ω|∑j=1,2F−[ϵj]δnF(E−(ϵj))+Θ|Ω|−2u2+Δ2×∑s=±Gs[ϵ3]δnF(Es(ϵ3))+F+[ϵ2]δnF(E+(ϵ2)).

Here, the functions $F_{\pm }\left(\epsilon \right)$ and $G_{\pm }\left(\epsilon \right)$ are defined by Appendix A, together with the dispersions
(9)$$\begin{array}{c} E_{\pm }\left(\epsilon \right)=\left|\sqrt{\epsilon^{2}-u^{2}}\pm \Delta \right|, \end{array}$$
and the frequency-dependent coefficients
(10)$$\begin{array}{cc} \epsilon_{1,2}= & \;\sqrt{u^{2}+{\left(\Delta \pm \sqrt{\frac{\Omega^{2}}{4}-u^{2}}\right)}^{2}},\\ \epsilon_{3}= & \;\sqrt{\frac{\Omega^{2}}{4}+\frac{u^{2}\Delta^{2}}{\Delta^{2}-\Omega^{2}/4}}. \end{array}$$

The optical conductivity is presented in Figure 1b, and it features two peaks, which represent hallmarks of this state. It is worth noticing that the weight of the conductivity peak at the lower frequency scales inversely with the amplitude of the order parameter Δ. For the technical details, consult Appendix A.

To further shed light on the presence of the two peaks in the optical conductivity, we analyze the corresponding DOS, with the form given in (Appendix A)
(11)$$\begin{array}{cc} \; & N_{BC}\left(E\right)=\frac{\left|E\right|}{6\pi \alpha^{2/3}}\frac{1}{\sqrt{E^{2}-u^{2}}}\left[{\left(\sqrt{E^{2}-u^{2}}-\Delta \right)}^{-\frac{1}{3}}\right.\\ \; & \times \left.\Theta \left(E^{2}-u^{2}-\Delta^{2}\right)+{\left(\sqrt{E^{2}-u^{2}}+\Delta \right)}^{-\frac{1}{3}}\Theta \left(E^{2}-u^{2}\right)\right] \end{array}$$
where we used that in this case, the dispersion in each of the four bands is given by $E\left(k\right)=\pm \sqrt{u^{2}+{\left(\alpha k^{3}\pm \Delta \right)}^{2}}$. This form of the band structure with two valence and two conduction bands, yielding the peaks in the DOS [Equation (Equation 11)], is therefore consistent with the form of optical conductivity in this phase, as shown in Figure 1b. We now analyze this observable in the remaining candidate phase for the paramagnetic metal, namely, the sCDW phase.

## 6. Charge Density Wave

The smectic charge-density wave yields a peculiar structure of the dispersion as it transforms under the two-dimensional *E* representation of the $D_{3}$ point group, with the vector realized in the valley space. As such, the sCDW mixes the valleys, with the bands given by
(12)$$E\left(k\right)=\pm \sqrt{u^{2}+\alpha^{2}k^{6}+\Delta^{2}\pm 2\Delta \sqrt{u^{2}+\alpha^{2}k^{6}\sin^{2}3\phi }},$$
and the corresponding band structure shown in Figure 3. In fact, the sCDW splits two cubic band touching points, with the vorticity ±3π, living in the two valleys, into two triplets of the points with the vorticity ±π, as explicitly shown in the Appendix A. Furthermore, the form of the low-energy DOS, featuring the linear-*E* behavior, shown in Figure 2c, is consistent with the splitting of the nodal points in the sCDW phase. See also Appendix A for the technical details.

We now compute the optical conductivity in the case when one of the components of this order parameter is nonzero, e.g., $\Gamma_{\rho 0}$, and for the sake of the clarity, since the term ∼*u* acts as a gap, we also set u=0. Furthermore, we take μ=0 and T=0, in the collisionless limit Ω≫Δ, with Δ as the order-parameter amplitude, to obtain $\sigma_{ij}=\sigma \delta_{ij}$, with
(13)Reσ(Ω)=σ0=38,
with the technical details shown in Appendix A. The identical results are obtained if instead we take $\Gamma_{\rho 3}$ to represent the sCDW order parameter, as expected from the corresponding commutation relations with the noninteracting Hamiltonian in Equation (Equation 1). In fact, the sCDW state can be thought of as a two-dimensional analogue of a correlation-driven nematic phase in a three-dimensional multi-Weyl semimetal [36].

This behavior of the optical conductivity is consistent with the splitting of the noninteracting band touching points with the vorticity of ±3π, into two triplets of the non-degenerate ones with the vorticity ±π. We emphasize that the obtained form of the optical conductivity singles out the sCDW phase as compared with the previously discussed valence-bond and bond-current states. Finally, the obtained mean-field value of the conductivity is expected to decrease when fluctuations are taken into account due to the scattering between quantum-critical fermionic and bosonic excitations [37,38].

## 7. Conclusions and Outlook

In this paper, we showed that the optical conductivity can distinguish possible paramagnetic metallic ground states yielding a superconducting order in RTG, as shown in Figure 1. In particular, the valence-bond and bond-current states are distinguishable by the number of peaks in the optical conductivity, while this observable in the sCDW order features a minimum with a value matching the noniniteracting one at the mean-field level, due to the splitting of the cubically dispersing nodal points at each valley. We emphasize that we here analyze only the real part of the high-frequency optical conductivity due to its universal features. On the other hand, its imaginary part can be trivially obtained by integration over frequency via the Kramers–Kronig relation Imσij(Ω)=−2Ωπ−1P∫0t/ℏdωReσij(ω)/(ω2−Ω2). However, its specific features are sensitive to (nonuniversal) microscopic details, such as the bandwidth, and hence, it does not provide a direct probe of the possible interaction-driven ground states in RTG.

The approach employed for the identification of the possible interaction-driven metallic ground states is quite general and can be applied to other vdW systems to distinguish possible correlated insulating and metallic states, as in the case of twisted bilayer graphene [27]. The same applies to the case of Bernal bilayer graphene (without twist) and other vdW materials, such as MoS${}_{2}$ and WSe${}_{2}$, where an analogous analysis can also be used to distinguish possible interaction-driven metallic ground states.

We here emphasize that our conclusions are based on the mean-field picture where the role of the fluctuations on the conductivity has been neglected. To account for the further corrections in the quantum-critical regime, we need to address the full quantum-critical theory describing the quantum-critical transition, as, for instance, in case of monolayer graphene [37], and we plan to investigate this problem in the future. Another open avenue emerging from this work concerns the features of the interaction-driven phases in vdW materials in the nonlinear transport [38]. Finally, our findings should stimulate future theoretical and experimental work on the out-of-plane optical response of the vdW materials [39], and particularly, the role of the electron–electron interactions in this respect.

## Figures and Tables

**Figure 1 nanomaterials-12-03727-f001:**
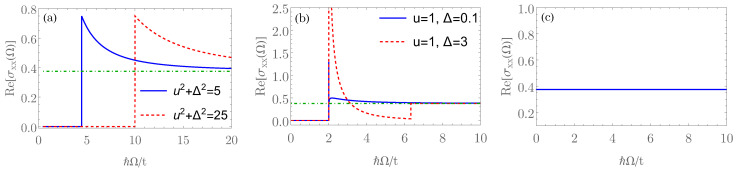
Real part of the optical conductivity (in units of $e^{2}/\hbar$) in the collisionless regime for: (**a**) the valence-bond order (VBO); (**b**) the bond-current order (BCO); (**c**) smectic charge-density wave (sCDW), respectively, given by Equations (Equation 6), (Equation 8), and (Equation 13), at the neutrality point μ=0 and at T=0. *u* and Δ are given in units of the bandwidth scale *t*, see also the discussion after Equation (Equation 1). In panels (**a**,**b**), the green dashed line corresponds to the universal optical conductivity for the non-interacting spinless RTG in the collisionless regime, $\sigma_{0}=3/8$.

**Figure 2 nanomaterials-12-03727-f002:**
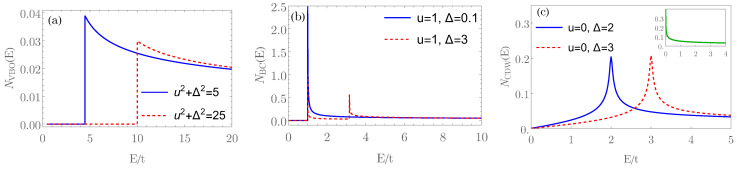
The density of states (DOS) in the candidate paramagnetic states. (**a**) valence-bond order (VBO); (**b**) bond-current order (BCO); (**c**) smectic charge-density wave (sCDW). The DOS in the three phases is given in Equations (Equation 7), (Equation 11), and Appendix A. The energy (*E*) is in units of the bandwidth (*t*), see also Figure 1. *u* and Δ are given in units of *t*. The inset in (**c**) corresponds to the DOS for the noninteracting case, where the low-energy DOS scales as ${\left|E\right|}^{-1/3}$.

**Figure 3 nanomaterials-12-03727-f003:**
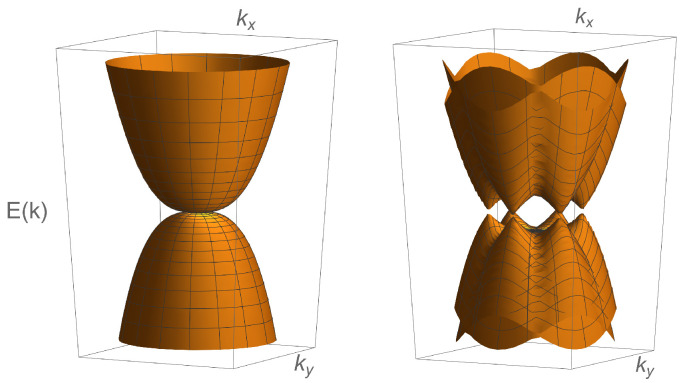
The band structure for the noninteracting Hamiltonian given by Equation (Equation 1) (left panel), where the pairs of degenerate bands corresponding to the same valley are superimposed. In the smectic charge-density wave with the order parameter $\Gamma_{\rho 0}$, ρ=1,2, the two cubic band touching points split into six points, with the triplets featuring opposite vorticities ±π (right panel). Notice that the smectic charge-density wave order parameter mixes the two valleys and, therefore, also breaks the original lattice translation symmetry, besides the rotational one. The rotational symmetry is, however, restored close to each of the new band touching points.

## Data Availability

Not applicable.

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
