# Peer review of "Optical Conductivity as a Probe of the Interaction-Driven Metal in Rhombohedral Trilayer Graphene"

_nanomaterials, 2022, doi:10.3390/nano12213727_

Round 1
Reviewer 1 Report
In this paper authors study the optical conductivity in rhombohedral trilayer graphene. They claim that features of the optical conductivity can distinguish different ground states.
Fig. 1 is explicitly for zero temperature. What about figs. 2 and 3? Are all the results reported in this paper for zero temperature? If yes, this should be clearly stated in the abstract.
In the caption of fig. 1 they write: The frequency is given in units of the bandwidth, t ∼ αk3 C, with kC as the momentum cutoff (hbar = 1 ). Can they also provide a numerical value in the international system of units?
In the caption of fig. 2 they write: The energy is given in units of the bandwidth. Can they also provide a numerical value in the international system of units?
These are minor revisions and if properly addressed the paper can be accepted for publication.
Reviewer 2 Report
In this paper, the authors show that the optical conductivity of rhombohedral trilayer graphene can distinguish the possible paramagnetic metallic ground states in this system by detailed calculation. It was found that the valence-bond and bond-current states are distinguishable by the number of peaks in the optical conductivity. In addition, the symmetry breaking charge-density wave state exhibits a minimal conductivity with the value independent of the amplitude of the order parameter, which corresponds to the splitting of the two cubic nodal points at the two valleys into two triplets of the band touching points.
The results obtained in this study are important for the understanding of the electron interactions in Quasi-two-dimensional graphene-based van der Waals heterostructures and can represent solid evidence of different candidate order-parameters for the paramagnetic metallic ground state. I therefore recommend its publication. Here are some suggestions which might improve the quality of their manuscript.
1) Whether other van der Waals heterostructures can also distinguish the paramagnetic metallic ground states by obtaining optical conductivity?
2) Whether the equations of the optical conductivity and density of states of rhombohedral trilayer graphene can be simplified to intuitively understand the phenomena in Figure 1-3 ?
Reviewer 3 Report
Strongly correlated states in van der Waals materials exhibit brand-new electronic states and can be controlled externally by twisting, doping, magnetic field, etc. Hence, it is of great interest in recent years to study electron-electron interaction in van der Waals materials both from theory and experiment aspects. Previous experimental studies show that rhombohedral graphene hosts various electron interaction phases and is an ideal platform for well-controlled tests of the many-body theory. However, one of the difficulties in understanding the electron interactions in these systems is distinguishing possible candidate ground states. In this manuscript, the authors show that optical conductivity can distinguish different candidates for a paramagnetic metallic ground state in rhombohedral graphene using low-energy Dirac theory. According to their theory, one can easily get useful information from optical conductivity related to the excitation spectrum. Specifically, only one peak is observable in the fully gapped valence-bond state, two peaks are observable in the bond-current state, and the rotational symmetry-breaking charge-density wave, therefore, exhibits a minimal conductivity. The predicted features represent the smoking gun signatures of different candidate order parameters for the paramagnetic metallic ground state in rhombohedral graphene. In my view, this manuscript is very useful in the field of van der Waals materials and electronics. I recommend it be published in Nanomaterials after minor revision.
1. In Figure 1, what is the green dashed line? What are the units of Ω In Figure 1b, can the author provide a zoom-in figure of the first peak or explain why the first peak is sharp and close to a vertical line? In the caption part and equation 6, the author should also include what are Ω, u, and Δ
2. In Figure 2, what are the units of E? In Figure 2c, what is the inset figure? There are no symbols and units. The author should explain it in the caption and main text part.
3. In Figure 3, the author should include units and values of k and E.
4. The authors calculated the real part of optical conductivity. Is there any way to calculate the imaginary part of optical conductivity so that it can be fully established?
